# Inspiring Tactics with the Improvement of Mitophagy and Redox Balance for the Development of Innovative Treatment against Polycystic Kidney Disease

**DOI:** 10.3390/biom14020207

**Published:** 2024-02-09

**Authors:** Moeka Nakashima, Naoko Suga, Yuka Ikeda, Sayuri Yoshikawa, Satoru Matsuda

**Affiliations:** Department of Food Science and Nutrition, Nara Women’s University, Kita-Uoya Nishimachi, Nara 630-8506, Japan

**Keywords:** polycystic kidney disease, chronic kidney disease, autophagy, mitophagy, mitochondria, hypoxia, adenosine monophosphate-activated protein kinase, gut microbiome

## Abstract

Polycystic kidney disease (PKD) is the most common genetic form of chronic kidney disease (CKD), and it involves the development of multiple kidney cysts. Not enough medical breakthroughs have been made against PKD, a condition which features regional hypoxia and activation of the hypoxia-inducible factor (HIF) pathway. The following pathology of CKD can severely instigate kidney damage and/or renal failure. Significant evidence verifies an imperative role for mitophagy in normal kidney physiology and the pathology of CKD and/or PKD. Mitophagy serves as important component of mitochondrial quality control by removing impaired/dysfunctional mitochondria from the cell to warrant redox homeostasis and sustain cell viability. Interestingly, treatment with the peroxisome proliferator-activated receptor-α (PPAR-α) agonist could reduce the pathology of PDK and might improve the renal function of the disease via the modulation of mitophagy, as well as the condition of gut microbiome. Suitable modulation of mitophagy might be a favorable tactic for the prevention and/or treatment of kidney diseases such as PKD and CKD.

## 1. Introduction

Polycystic kidney disease (PKD) is the most familiar genetic type of CKD. The condition is characterized by the development of multiple kidney cysts, and it could subsequently instigate kidney damage and/or renal failure [1]. This disease may be triggered by mutations in *PKD1* or *PKD2* genes, which encode the integral membrane proteins polycystin-1 and polycystin-2, respectively [2]. Interestingly, these PKD proteins are associated with decreased autophagy [2,3]. Autophagy has been found to be impaired in the epithelial cells of the kidneys in animal models of PKD, as well as in patients with PKD, suggesting that this impairment might contribute to the development and/or progression of PKD [4,5]. In addition, several agents, such as rapamycin, could protect against PKD, which might restore the autophagy in animal models [6]. Mechanistically, aberrant activation of the mammalian target of rapamycin (mTOR), a target molecule of rapamycin, has been shown to be linked to the impaired autophagy as well as the pathology of PKD [7]. Cyst enlargement in PKD kidneys may result in restricted areas of hypoxia [8]. Hypoxic stimuli may then increase the hypoxia-inducible factor-1α (HIF-1α) protein by preventing its degradation by the proteasome. Under hypoxic conditions, the phosphatidylinositol 3-kinase (PI3K) and the mTOR pathway might activate the expression of HIF-1α [9]. (Figure 1) Hypoxia-related events have been revealed to be linked with cyst formation [10,11]. HIF-1α has been found to be also highly expressed in dendritic cells, and its expression is relatively higher in radicular cysts than in odontogenic tumors [12]. In PKD, HIF-1α may not disturb initial cyst formation, but it is important for cyst progression and expansion in later stages of the disease [13]. Conversely, it has also been shown that HIF-1α inhibition could reduce cystic growth [14]. Signaling pathways being related to the activation of HIF-1α during hypoxia could be contributing to cyst expansion in PKD [15]. Reactive oxygen species (ROS) have been also shown to stimulate cyst development in PKD. In addition, several tissues of PKD may exhibit elevated ROS levels that are positively interrelated with disease severity [15,16]. Peroxidation of phospholipids in animal and human kidneys may be caused by high amounts of ROS [17,18]. Cultured renal cysts and MDCK cell cysts in a three-dimensional setup have confirmed a relationship between lipid peroxidation and increased cyst size [14,19]. It is well-known that damaged mitochondria could induce the generation of ROS and may bring about an increase in membrane lipid peroxidation. Therefore, autophagy, mitochondria, hypoxia and/or ROS might be important administrators in the progression in PKD. Undoubtedly, these hypotheses need further investigation. In addition, despite remarkable efforts to clarify all features of PKD through wide-ranging translational research, there is still an unmet clinical requirement for biomarkers and/or prognosticators that may possibly predict the speed of disease progression [20,21,22]. A better comprehending of the pathophysiology of cystic expansion may lead to the advancement of potential therapies to slow cyst development and/or expansion. The development of innovative treatments that may act synergistically or have fewer side effects might considerably improve the treatment consequences.

## 2. Autophagy/Mitophagy and Redox Imbalance in the Homeostasis of Kidney Cells

Substantial evidence has supported an imperative role for autophagy in kidney pathophysiology. Autophagy is firmly regulated to support cells to get used to and/or decrease cellular stress. Some studies have emphasized an intricate signaling system that could detect alterations in energy and/or nutrient condition to either activate or prevent autophagy. Intracellular stresses, which can be brought from ROS, hypoxia, stress of endoplasmic reticulum, several DNA damages and/or inflammatory immune signaling have been revealed as potential stimulators of autophagy [23]. Interestingly, autophagy has been defined as a HIF-1α-dependent response [24]. Autophagy describes the process by which cytoplasmic materials, including organelles, access the lysosomes for hydrolytic degeneration [25], which is also a course of cell repair that may frequently convey the apoptosis termed “self-killing” of cells [26]. Dying cells may frequently exhibit an accumulation of autophagosomes and hence adopt a morphology known as autophagic cell death [26]. Consequently, autophagic cell death might cause cell death with autophagy rather than cell death by autophagy. Hypoxia can regulate the mTOR complex 1 (mTORC1) [27]. Therefore, hypoxia and/or mTOR signaling may be modulators of autophagy [27]. (Figure 1) Mitochondria are particularly sensitive to hypoxa, which might result in both functional and morphological impairments. Mitophagy is an arrangement of autophagy that eliminates surplus mitochondria, facilitates reconstruction of mitochondria, and prevents the accumulation of impaired mitochondria [28]. Therefore, mitophagy might be a key mechanism for preserving the quality of mitochondria by eliminating damaged mitochondria. In response to hypoxia, the PTEN-induced putative kinase 1 (PINK1) may be activated as a regulator of mitophagy, confirming the suitable functioning of the total mitochondrial network [29]. Various stressors, such as hypoxia, ischemia, ageing, and oxidative stress, may lead to an increase in ROS and damages to mitochondria, which may trigger the PINK1 mediated mitophagy [30]. (Figure 2) There is evidence for weakened mitophagy in the renal cells of diabetic mice with reduced expressions of mitochondrial PINK1 [31]. A working mitophagy system may act as a scavenger of damaged mitochondria, and thereby maintain a decent mitochondrial homeostasis.

Mitophagy is principally facilitated by microtubule-associated protein 1 light chain 3 (LC3)-linked receptors. Ubiquitin-dependent mitophagy may include the mitochondrial serine/threonine protein kinase PINK1 and E3 ubiquitin protein ligase Parkin; it also may include the Parkin/PINK1 pathways [32]. Conclusions of the experiment in primary human renal epithelial cells have demonstrated that mitochondrial quality control could be disturbed by mitophagy mediated via PINK/Parkin signaling [33]. PINK1 accumulates on the outer mitochondrial membrane (OMM) after loss of mitochondrial membrane potential, where it recruits and then phosphorylates Parkin to add phosphor-ubiquitin chains on OMM proteins. Interestingly, mitophagy could inhibit oxidative stress via the upregulation of the PINK1-parkin pathway, which could delay kidney senescence in mice [34]. Autophagy receptors, including optineurin, calcium-binding and coiled-coil domain-containing protein 2, also called nuclear dot protein 52 kDa (NDP52), which comprise both ubiquitin binding domains and LC3-interacting regions, could link the ubiquitylated mitochondria to LC3-associated membranes for appropriation [35]. PINK1-mediated phosphorylation of ubiquitin can employ optineurin and/or NDP52 to induce mitophagy without Parkin. By attaching to LC3 at their cytosolic N-terminus, mitophagy receptors could connect impaired mitochondria directly to autophagosomes. (Figure 2) After ubiquitination, impaired mitochondria might be consequently recognized by adapter proteins to be eaten by autophagosomes. Too much mitophagy might result in cellular energy depletion. Therefore, mitophagy may positively or negatively regulate apoptosis, which is a double-edged sword in the pathogenesis of several diseases. For example, a high or low level of mitophagy activity may occasionally induce podocyte apoptosis, which is the collective pathological base for the progression of several kidney diseases [36]. In general, mitophagy may be induced as a protection mechanism for keeping a population of well mitochondria and thus safeguarding cell survival. Although mitophagy may be dispensable for kidney development [37], mitophagy seems to be essential for maintaining kidney integrity and normal physiology in adult kidney cells [38]. Clearance of damaged mitochondria via mitophagy is valuable to the protective effect of impaired kidney cells [39].

## 3. Autophagy/Mitophagy Involved in the Pathogenesis of Several Kidney Diseases, including Polycystic Kidney

Collecting evidence relates the impaired mitophagy with disease pathogenesis/progression in several pathological situations, including kidney diseases [40]. Acute kidney injury (AKI) may be categorized by a rapid weakening of kidney function, which typically results from renal ischemia, sepsis, and nephrotoxic agents [41]. Mitophagy induction might act as a mutual mechanism to kidney tubular cell protection in many models of AKI [42]. The mitophagy-mediated removal of injured mitochondria might inhibit excessive ROS accumulation, as well as prevent the release of damage-associated molecular arrays which might indorse inflammation during AKI. As renal tissue has massive mitochondrial content, mitophagy and/or mitochondrial biogenesis may be critical to overwhelming stressful illnesses, including AKI [42,43]. Mitophagy might enable compromised cells to persist during kidney interstitial fibrosis, which is a feature of maladaptive restoration in the transition from AKI to chronic kidney disease (CKD) [44]. Mitophagy being induced in distal tubules and pericytes could protect against renal interstitial fibrosis by suppressing the inflammasome of the tumor growth factor β (TGFβ) and the NLR family pyrin domain containing 3 (NLRP3) signaling [45]. Therefore, mitophagy might be a pharmacological target for the management of interstitial fibrosis in kidneys, particularly in regard to offering new concepts for more efficient anti-fibrosis and delaying the development of CKD [46]. It has been shown that mitophagy activation may protect against renal fibrosis via the downregulation of TGF-β1/Smad signaling, improving mitochondrial fitness and alleviating inflammatory infiltration in kidneys [47]. Focal segmental glomerulosclerosis may be one of the fibrotic diseases in kidneys that is characterized by glomerular lesions with podocytes [48]. It has been revealed that podocyte mitophagy could have an imperative role in the development of the focal segmental glomerulosclerosis [48]. In addition, modifications of the apolipoprotein L1 (*APOL1*) gene have known links with the focal segmental glomerulosclerosis, which may affect endosomal trafficking and/or block mitophagic flux, eventually leading to podocyte injury [49]. Therefore, podocyte mitophagy could counteract the development of the focal segmental glomerulosclerosis [50]. A decrease in podocyte mitophagy may underlie the conceivable progression of podocytopathies, including the focal segmental glomerulosclerosis [51]. Interestingly, activation of mTORC1 has been detected in glomeruli from patients with the focal segmental glomerulosclerosis [52]. Hyperglycemia may inhibit mitophagy in kidney tubules of diabetic patients with diabetes mellitus [53]. It seems that mitophagy has been impaired in the diabetic kidneys of patients with diabetic kidney disease [53]. Defective mitophagy induced by high glucose levels may accelerate the senescence of tubular cells [54]. Treatment with the mitochondria-targeted antioxidant may ameliorate tubular injury in diabetic mice by restoring mitophagy, which might be mediated by an elevation in PINK1 expression provoked by nuclear factor erythroid 2-related factor 2 [53,54]. Therefore, mitophagy in kidney tubules might be helpful for diabetic kidney disease.

Autosomal-dominant polycystic kidney disease is a popular heritable human disease featuring the final development of renal failure, which is caused by mutations in either *PKD1* or *PKD2* genes. These gene product polycystins (PC1 and PC2) might play crucial roles in ensuring proper mitophagic processes. In fact, PKD is one of the most common ciliopathies that may be associated with decreased mitophagy [3,55]. The existence of cilia may to be essential in the activation of mitophagy [56]. Accordingly, impairment of mitophagy might suppress ciliagenesis [56]. At present, effective treatment seems to be lacking, while inhibition of mTOR may slow cyst expansion in animal models. Several agents that may protect against PKD in animal models could also restore mitophagy, suggesting that mitophagy might be associated with a pathogenic role in PKD [6,57]. Interestingly, abnormal mTOR activation could be connected to the impaired mitophagy and/or defective cilia in PKD [7,58].

## 4. Autophagy/Mitophagy as a Target of Treatment against Polycystic Kidney Disease

A number of studies have emphasized the dysregulation of mitophagy in PKD, representing both the augmented and diminished activities of mitophagy. Impaired mitophagy could lead to the accumulation of damaged mitochondria and cyst formation, while increased mitophagy might exacerbate the cyst growth. Therefore, treatment with chemical autophagy activators, including mTOR-dependent rapamycin, could slightly but noticeably attenuate cyst formation and repair the kidney function [59]. It has been shown that the *PKD1* gene, which encodes the polycystin-1 (PC1) protein, is responsible for 85% of cases of autosomal-dominant polycystic kidney disease [60]. The PC1 could regulate the function of calcium-dependent calpain proteases, which may preserve lysosomal integrity [61]. In addition, failure of PC1 function might be associated with the development of renal cysts and/or weakened kidney function. The polycystin-2 (PC2) constructs a complex with beclin-1, which might exert a key role involved in the formation of autophagic vacuole [62,63]. Therefore, PC2 is a critical mediator of mitophagy initiation [63,64]. Basal autophagy may be boosted in PC1-deficient cells, implying that PC1 might promote autophagic cell survival [65]. Interestingly, it has been shown that steviol, a metabolite of the sweetening chemical compound stevioside, could decelerate the cyst development in renal epithelial cells by increasing PC1 expression and by stimulating lysosomal degradation of β-catenin in animal models of PKD [66]. In addition, the stevioside metabolite could enhance the autophagy via the stimulation of an adenosine monophosphate-activated protein kinase (AMPK) pathway [67]. Trehalose is also a natural, nonreducing disaccharide comprising two glucose molecules linked by an α, α-1,1-glucosidic bond that has been shown to enhance autophagy. Trehalose is found in microorganisms, plants, insects, and invertebrates, but not in mammals [68]. Trehalose could defend the integrity of cells against several damages, including oxidation and/or hypoxia, by decreasing protein denaturation via the protein–trehalose interaction [69], which has been utilized in the preservation of food but also applied to deal with medical diseases because of its ability to augment autophagy. In adults, oral trehalose supplements may improve the vascular function by increasing redox balance [70]. Furthermore, trehalose may exert cytoprotective effects in podocytes and in proximal tubular cells by inducing autophagy [71,72].

In general, autophagy could be stimulated by nutrient and/or energy deprivation, which may be regulated by signaling pathways with AMPK and/or mTOR. The mTOR could construct the rapamycin-sensitive mTORC1 and the rapamycin-insensitive mTORC2, which might be key regulators of autophagy [73]. Under nutrient-rich conditions, however, mTORC1 suppresses autophagy by phosphorylating the unc-51-like autophagy activating kinase 1 (ULK1) and the autophagy related 13 (ATG13) [74]. Active mTORC1 could also stimulate ribosome biogenesis and mRNA translation by phosphorylating p70 ribosomal protein S6 kinase (p70S6K), as well as the eukaryotic translation initiation factor 4E-binding protein 1 (4E-BP1) [75]. Well-known inhibitors of mTORC1 include rapamycin and rapamycin analogues [76]. In hunger situations, mTORC1 is repressed and detaches from the ULK1–ULK2 complex, permitting ULK1 to be triggered by AMPK to induce autophagy [77]. Sirtuins, from SIRT1 to SIRT7, are nicotinamide adenine dinucleotide (NAD^+^)-dependent class III histone deacetylases. In the conditions of energy depletion, SIRT1 Is stimulated by increased NAD^+^ levels. Dynamic SIRT1 could activate autophagy through the deacetylation of ATG proteins and/or of the transcription factor forkhead box protein O1 (FOXO1) and forkhead box protein O3a (FOXO3a), which can transactivate autophagy genes [78,79,80]. Moreover, crosstalks of Sirtuin-1 (SIRT1) with the mTOR and AMPK pathway may control cell survival and/or autophagy by adjusting diverse mechanisms involved in energy metabolism [81,82]. Interestingly, resveratrol could appropriately activate SIRT1 [83], which might attenuate the oxidative stress and/or the mitochondrial dysfunction partly via the mitophagy [84]. SIRT1 is upregulated in the autosomal-dominant polycystic kidney disease and accelerates disease progression by deacetylating the p53 tumor suppressor. Niacinamide, also known as nicotinamide, is a dietary supplement and a non-competitive inhibitor of sirtuins that can reduce proliferation and augment apoptosis of cystic epithelial cells by preventing the deacetylation of p53 [85]. Long-term calorie restriction could restore the autophagic activity via the activation of SIRT1, which may protect from mitochondrial damages in the kidney induced by oxidative stress [86]. In addition, calorie restriction might also enhance the autophagy in podocytes and in proximal tubules [87]. Metformin could affect cells via the activation of AMPK [88,89], which is a key regulator of several pathways involved in energy, glucose, and lipid metabolism, as mentioned above. The blockade of AMPK signaling could considerably influence the efficiency of metformin for the type-2 diabetes mellitus and/or atherosclerosis [90,91]. Also, metformin plays roles in altering the pathogenesis of diseases by restoring the redox balance and influencing mitochondrial function [92,93]. Moreover, metformin can improve mitochondrial bioenergetics by increasing autophagy [94,95].

## 5. Another Tactic Regarding the Alteration of Gut Microbiome for the Treatment of Polycystic Kidney

Decreased fatty acid oxidation and/or a dysregulated lipid metabolism have been recognized as key PKD features [96]. Remarkably, treatment with the peroxisome proliferator-activated receptor-α (PPAR-α) agonist could enhance the fatty acid oxidation and reduce cystic disease in PKD models [97]. Peroxisome proliferator-activated receptors (PPARs) are categorized as members of the nuclear receptor family of transcription factors, which may be activated by several fatty acids and their derivatives [98]. It is well-known that some fatty acids may modulate the autophagy. In addition, gut microbiota may play crucial roles in some pathological processes through controlling several metabolic factors, including certain fatty acids [99]. Therefore, reciprocal interactions have been observed between PPARs and the gut microbiota in both healthy and diseased conditions, indicating that the nuclear receptors might be good targets for treatment of various diseases through the crosstalk with the gut microbiota [99]. The discovery of the gut–kidney axis may also establish the relationship between the disruption of gut homeostasis and CKD onset and/or progression, which may be regulated by the gut microbiota and/or immune cells [100,101]. Autosomal-dominant polycystic kidney disease may be the prominent cause of inherited kidney disease, with significant contributions to CKD [102]. Hence, treatment against CKD might be also beneficial for the treatment of polycystic kidney disease. The relationship between CKD and gut dysbiosis is also bidirectional. For example, gut-derived metabolites and/or toxins could influence the progression of CKD, and the uremic situation might also affect the gut microbiota [103]. Intestinal dysbiosis may contribute to the compromised intestinal barrier function, which could facilitate the translocation of uremic metabolites from the gut to the blood, contributing to the elevation of oxidative stress and CKD progression [104]. With increased permeability of the colon-intestinal epithelium, pathogens and/or antigens could come into systemic circulation, which may also lead to CKD progression. Remarkably, it has been shown that *Bifidobacterium* and *Lactobacilli* sp. in the gut may be negatively correlated with CKD progression and long-term survival [105]. In addition, the presence of *Roseburia*, *Faecalibacterium prausnitzii*, and/or *Prevotella* may be also negatively correlated with uremic toxin accumulation and disease progression [105]. The colon–intestinal tract might be protected by a huge number of immune cells and structures for the appropriate homeostasis. Several infections may be common factors in critical exacerbations of CKD, which may be resulting from immune and inflammatory responses [106,107]. Alterations in the gut microbiota might be sometimes beneficial for the CKD regression via the metabolic changes, immune modification, and/or reduced inflammation. Therefore, using prebiotics, probiotics, and/or fecal microbiota transplantation (FMT) to regulate the gut ecology may alleviate oxidative stress as well as improve kidney function [108]. (Figure 3) Remarkably, it has been reported that oral supplementation of short chain fatty acid may amend kidney functions in rats, possibly by enhancing autophagy/mitophagy via the AMPK/mTOR pathway [109]. Lastly, there is some evidence that the gut microbiome is possibly altered in patients with CKD and polycystic kidney disease [110].

## 6. Future Perspectives

Given the vital role of mitophagy in the development of various kidney diseases, suitable modulation of mitophagy might be a promising tactic for the prevention and/or treatment of kidney diseases, including polycystic kidney disease. In addition, pharmacological modulation of autophagy has been useful in some experimental models of AKI and chronic kidney injury (CKI). However, the precise advantageous function of mitophagy in kidney cells remains controversial. Although many signaling pathways may participate in the regulation of mitophagy in various organs, their detailed mechanisms also remain mostly unknown and/or complicated. Furthermore, signaling of mitophagy might interact with other cellular routes to influence the development of other renal diseases. Among them, CKD might be a major public health concern affecting more than 10% of the global population [111]. In general, dietary restrictions have been used to treat the CKD. Additionally, interventions such as synbiotics, prebiotics, and probiotics may improve the balance of the gut microbiota and enhance gut barrier function, which may also contribute to the amelioration of the kidney function.

Against the polycystic kidney diseases, however, those interventions with probiotics/prebiotics may be somewhat inadequate in regard to the improvement of the kidney function [109,110]. What are the additional factors/signaling required? Firstly, some antioxidants and/or redox balance might be valuable, as mentioned in the previous section. For example, significant studies have suggested that metformin exerts its favorable effect by various mechanisms, including affecting mitochondrial function, restoring of redox balance, and modulating the gut microbiome, which may be apart from the AMPK-dependent mechanism [112]. Preceding studies have shown the inhibition of nuclear factor (erythroid-derived 2)-like 2 (Nrf2), which is involved in regulating the expression of antioxidant proteins such as heme-oxygenase 1(HO-1) and/or catalase in animal models of chronic kidney disease (CKD), and it also may increase the inflammation and oxidative stress [113]. Resveratrol may be favorable for saving the redox balance, which is a polyphenolic chemical compound isolated from *Veratrum grandiflorum* with a diversity of biological activities, including antioxidant and/or anti-inflammatory properties [114]. In fact, resveratrol administration could inhibit HIF-1α expression, which may reduce the production of hypoxia-induced reactive oxygen species (ROS) [115]. Secondly, some factors involved in the tryptophan and kynurenine pathway might also be conceivable, which has been shown to be associated with immune-related diseases [116]. Several studies have described abnormal expression of the genes-encoding factors of the immune response in autosomal-dominant polycystic kidney disease, in which the immune system and/or infiltration of immune cells may be mostly stimulated [117]. The events associated with inflammation and/or activation of the immune system might promote the pathology of PKD [118]. Therefore, both polycystic kidney and autosomal-dominant polycystic kidney disease may be also categorized as immune-related diseases. Shaped by repetitive inflammatory conditions, an “engram” might commit to a mild progression of several immune-related diseases [116]. If that is the case in PKD, a certain “engram” modulation with the modification of gut microbiome might be beneficial for a superior treatment tactic against PKD [116]. (Figure 3) Additionally, a substantial proportion of PKD patients may experience hypertension prior to kidney dysfunction [119]. Remarkably, a significant proportion of PKD patients with normal kidney function may also progress to hypertension prior to the development of polycysts, suggesting that PKD2 channels can regulate blood pressure, probably through an extrarenal mechanism [120]. Hypertension is an important prognosticator of the disease progression, which might be the most frequent cause of death in patients with autosomal-dominant polycystic kidney disease. Angiotensin-converting enzyme inhibitors may be used as therapeutic agents in the treatment of hypertension in autosomal-dominant polycystic kidney disease [121]. The therapeutic benefit of using angiotensin-converting enzyme inhibitors may contribute to treating hypertension, as well as to diminishing renal cyst growth in the autosomal-dominant polycystic kidney disease [122].

## 7. Conclusions

Mitophagy may play an imperative role in the pathology of various kidney diseases, including PKD, CKD and/or AKI. Appropriate mitophagy might be firmly regulated to enable cells to lessen cellular stress, probably via sustaining the redox balance. Prior research has indicated that those kidney diseases might be also related with gut dysbiosis, which may lead to the development and/or progression of kidney diseases. Superior modification of mitophagy in kidneys via the modulation of the gut microbiome may contribute to the development of prevention/treatment tactic for several kidney diseases, including PKD.

## Figures and Tables

**Figure 1 biomolecules-14-00207-f001:**
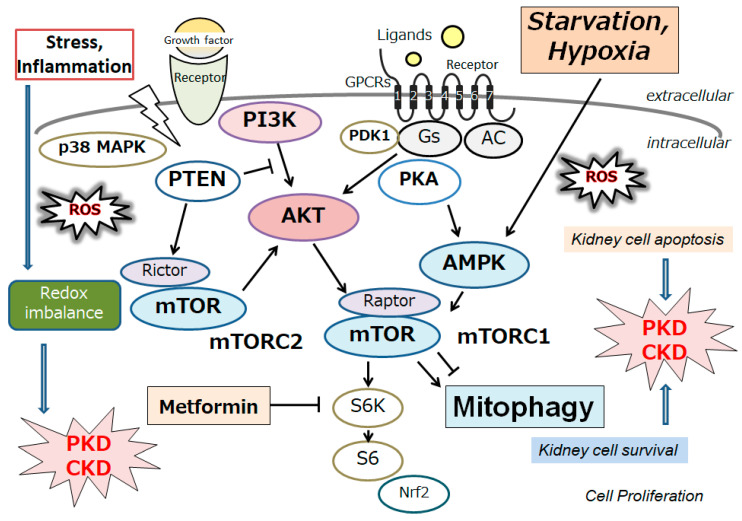
Schematic representation of the relevant signaling pathway potentially being involved in the pathogenesis of polycystic kidney disease (PKD) and/or chronic kidney disease (CKD). Several modulator molecules linked to the PI3K/AKT/mTOR/mTORC1 signaling pathway are demonstrated. Examples of compound metformin, as well as hypoxia and/or starvation, known to act on the AMPK/mTOR and/or mitophagy signaling, are also shown. Arrowhead indicates stimulation, whereas hammerhead shows inhibition. Note that several important activities, such as cytokine-induction and/or inflammatory reactions, have been omitted for clarity. Abbreviation: mTOR, mammalian/mechanistic target of rapamycin; PI3K, phosphoinositide-3 kinase; ROS, reactive oxygen species.

**Figure 2 biomolecules-14-00207-f002:**
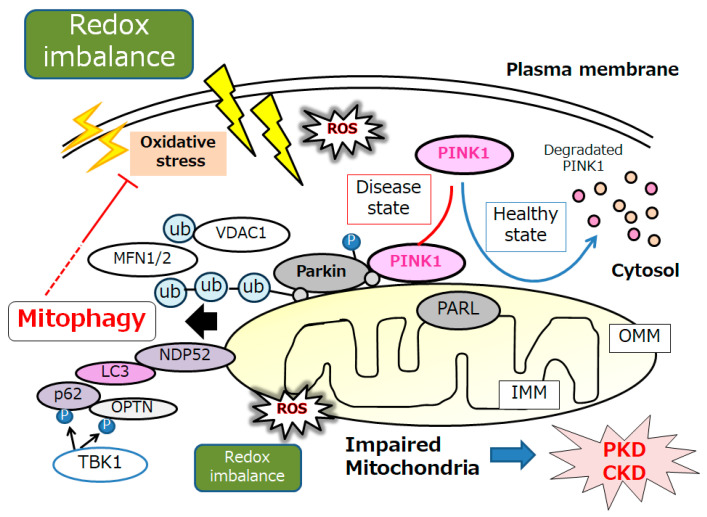
An illustrative representation and overview of PINK1, Parkin, and related molecules in the regulatory pathway for mitophagy. Under the healthy and steady state of cells, PINK1 is despoiled within the surface of mitochondria, which may be reduced by mitochondrial damage due to oxidative stress and/or redox imbalance, resulting in PINK1 and Parkin increases in the outer membrane of mitochondria. Mainly, the PINK1 could phosphorylate ubiquitin to activate the ubiquitin ligase activity for Parkin, where the Parkin is expected to be phosphorylated and ubiquitinated, resulting in the induction of mitophagy. OMM, outer mitochondrial membrane; IMM, inner mitochondrial membrane; MARK2, microtubule affinity regulating kinase 2; MFN1, mitofusin 1; MFN2, mitofusin 2; NDP52, nuclear dot protein 52; PARL, presenilin-associated rhomboid-like; OPTN, optineurin; PINK1, PTEN-induced kinase 1; ROS, reactive oxygen species; VDAC1, voltage-dependent anion channel 1; Ub, ubiquitin.

**Figure 3 biomolecules-14-00207-f003:**
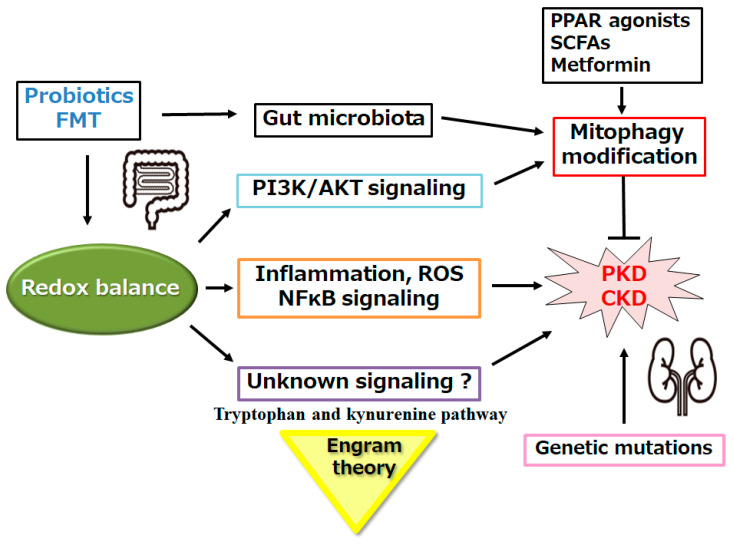
Schematic demonstration of the potential strategies against the pathogenesis of various kidney diseases, including polycystic kidney disease (PKD) and chronic kidney disease (CKD). Some kinds of probiotics and/or fecal microbiota transplantation (FMT) might assist the alteration of the gut microbiome for the modification of mitophagy, which might be advantageous in the treatment of several kidney diseases, including polycystic kidney disease (PKD) and chronic kidney disease (CKD). Note that some important activities, such as autophagy initiation, inflammatory reaction, and reactive oxygen species (ROS) production, have been misplaced for clarity. “?” represents author speculation. PPAR: peroxisome proliferator-activated receptor. SCFAs: short chain fatty acids.

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
