# Peer review of "Inspiring Tactics with the Improvement of Mitophagy and Redox Balance for the Development of Innovative Treatment against Polycystic Kidney Disease"

_biomolecules, 2024, doi:10.3390/biom14020207_

Round 1

Reviewer 1 Report

Comments and Suggestions for Authors

Section 5 doesn’t seem in line with the title of the article. Either should trim or remove this section or expand on the role of mitophagy and the gut microbiome.

Additional comments:

Line 18: “The mitophagy served …” should be changed to “Mitophagy serves …”

Line 66: remove the word “which” in front of “could subsequently …”

Line 68: “this …” should be changed to “these …”

Line 79: remove the word “the” in “front of hypoxia inducible factor-1α …”

Line 81: remove the comma in front of “it is important …”

Line 82: remove “the” in front of “cystic …”

Line 86: remove the word “probably” from this sentence.

Line 121: need a word between “might” and “cell …” Maybe the word cause?

Line 124: The phrase “affects to eliminating surplus mitochondria, facilitating the reconstruction of mitochondria, and preventing impaired mitochondria accumulation” should be changed to “… eliminates surplus mitochondria, facilitates mitochondrial reconstruction, and prevents the accumulation of impaired mitochondria.”

Line 126: change the word “saving …” to “preserving …”

Line 129: I think it would be good to expand on the “various stressors” as they, according to your review, could be an important driver of PKD.

Line 130: I would change “are evidences …” to “is evidence …”

Lines 156-157: not sure of the relevance of mitophagy in glomeruli vs. proximal tubules. This sentence should be removed or expanded.

Line 172: “evidences …” should be singular, “evidence …”

Line 200: which “mitochondrial targeted antioxidant?” Do you mean mitochondrial antioxidants in general or specific ones?  

Line 209: I would suggest changing “mitophagy flaws …” to “impairment of mitophagy …”

Lines 210-211: saying that “agents that may slightly protect against PKD in animal models could also restore mitophagy” doesn’t seem like a strong endorsement of your thesis presented here.  

Lines 255-257: The phrases “resveratrol could activate SIRT1, which might attenuate the oxidative stress and/or the mitochondrial dysfunction” and “SIRT1 is upregulated in the autosomal dominant polycystic kidney disease and accelerates disease progression …” seem contradictory. Is upregulation of SIRT1 good or bad?

Line 258: you should expand on the use niacinamide here.

Lines 266-267: this sentence should be moved to the next section where gut microbiota is discussed.

Line 311: change “promising tactics …” to “a promising tactic …”

Comments on the Quality of English Language

English is good, but there are few words and phrases that aren't quite right and are mentioned in the "suggestions for authors" section.

Author Response

For Reviewer1

Section 5 doesn’t seem in line with the title of the article. Either should trim or remove this section or expand on the role of mitophagy and the gut microbiome.

Accordingl to this suggestion, the Section 5 has been expandeded on the role of mitophagy and the gut microbiome. In addition, the title of the section 5 is altered.

Additional comments:

Line 18: “The mitophagy served …” should be changed to “Mitophagy serves …”

Corrected, thank you.

Line 66: remove the word “which” in front of “could subsequently …”

 Removed.

Line 68: “this …” should be changed to “these …”

Corrected.

Line 79: remove the word “the” in “front of hypoxia inducible factor-1α …”

Removed.

Line 81: remove the comma in front of “it is important …”

Removed.

Line 82: remove “the” in front of “cystic …”

Removed.

Line 86: remove the word “probably” from this sentence.

Removed

Line 121: need a word between “might” and “cell …” Maybe the word cause?

“cause” has been inserted, thank you.

Line 124: The phrase “affects to eliminating surplus mitochondria, facilitating the reconstruction of mitochondria, and preventing impaired mitochondria accumulation” should be changed to “… eliminates surplus mitochondria, facilitates mitochondrial reconstruction, and prevents the accumulation of impaired mitochondria.”

Improved, accordingly. Thank you so much.

Line 126: change the word “saving …” to “preserving …”

Altered.

Line 129: I think it would be good to expand on the “various stressors” as they, according to your review, could be an important driver of PKD.

Expanded, accordingly. Thank you.

Line 130: I would change “are evidences …” to “is evidence …”

Improved, thank you.

Lines 156-157: not sure of the relevance of mitophagy in glomeruli vs. proximal tubules. This sentence should be removed or expanded.

Accordingly, the description has been altered. Thank you so much.

Line 172: “evidences …” should be singular, “evidence …”

Corrected, thank you.

Line 200: which “mitochondrial targeted antioxidant?” Do you mean mitochondrial antioxidants in general or specific ones?  

We mean it includes specific one such as MitoQ that is described in reference 30.

Line 209: I would suggest changing “mitophagy flaws …” to “impairment of mitophagy …”

Improved, accordingly. Thank you.

Lines 210-211: saying that “agents that may slightly protect against PKD in animal models could also restore mitophagy” doesn’t seem like a strong endorsement of your thesis presented here.  

Removed the word “slightly”. Thank you.

Lines 255-257: The phrases “resveratrol could activate SIRT1, which might attenuate the oxidative stress and/or the mitochondrial dysfunction” and “SIRT1 is upregulated in the autosomal dominant polycystic kidney disease and accelerates disease progression …” seem contradictory. Is upregulation of SIRT1 good or bad?

Probably, it may depend on the expression level of SIRT1. I suppose that appropriate expression of SIRT1 could properly activate the autophagy which might be beneficial for the improvement of PKD. High level of SIRT1 expression, however, would be bad as described in the manuscript. Too much autophagy may also lead to cell apoptosis instead of cell protection.

Line 258: you should expand on the use niacinamide here.

Expanded, accordingly. Thank you.

Lines 266-267: this sentence should be moved to the next section where gut microbiota is discussed.

This sentence have been altered with the different references, accordingly. Thanks.

Line 311: change “promising tactics …” to “a promising tactic …”

Corrected, thank you.

Comments on the Quality of English Language

English is good, but there are few words and phrases that aren't quite right and are mentioned in the "suggestions for authors" section.

Improved as much as possible. Thank you so much.

Reviewer 2 Report

Comments and Suggestions for Authors

This review addresses the involvement of mitophagy and redox state in kidney diseases such as polycystic kidney disease (PKD) and chronic kidney disease (CKD) by discussing how modulation of mitophagy could represent a favorable strategy for the prevention and/or treatment of these kidney diseases.

Although the topic is remarkably interesting and overall adequately addressed, there are however some critical points that require a major revision of the review

Major points:

Both abstract and introduction omit the purpose of the review

The first sentence of the abstract should be eliminated or postponed after the next one.

Overall the review requires an extensive linguistic revision:

to give some examples, widespread and inappropriate use of the modal verb "could" or “might” was made in the manuscript; below are reported just some of the sentences in which "could" or “might” were used incorrectly:

line 67: delete “could”

line 117: delete “might”

line 121: change “could” with “can”

line 150: change “may” instead of “might”

line 224: delete “could”

line 274: delete “could” or change with may

Other examples of incorrect or obscure sentences:

Line 302: change and clarify the meaning of “inhibition against the pathogenesis”

Lines 336-337: clarify the meaning of “If that is the case, “engram” modulation with the modification of gut microbiome might be beneficial for the superior treatment tactic

All the figures need to be extensively modified because they are too generic, unclear and poorly detailed both in the diagrams and in the legends and what is "omitted for clarity" does not actually make the figure understandable.

Comments on the Quality of English Language

Extensive editing of English language required

Author Response

For Reviewer2

This review addresses the involvement of mitophagy and redox state in kidney diseases such as polycystic kidney disease (PKD) and chronic kidney disease (CKD) by discussing how modulation of mitophagy could represent a favorable strategy for the prevention and/or treatment of these kidney diseases.

 Although the topic is remarkably interesting and overall adequately addressed, there are however some critical points that require a major revision of the review

Major points:

Both abstract and introduction omit the purpose of the review

The first sentence of the abstract should be eliminated or postponed after the next one.

According to this suggestion, the first part of the abstract and introduction have been improved.

Overall the review requires an extensive linguistic revision:

to give some examples, widespread and inappropriate use of the modal verb "could" or “might” was made in the manuscript; below are reported just some of the sentences in which "could" or “might” were used incorrectly:

According to this suggestion, we have gone over the text/abstract and amended typos and grammatical errors as much as possible to improve the manuscript more helpful to the readers.

line 67: delete “could”

Deleted, thank you.

line 117: delete “might”

Deleted, thank you.

line 121: change “could” with “can”

Changed, thank you.

line 150: change “may” instead of “might”

Changed, thank you.

line 224: delete “could”

Deleted, thank you.

line 274: delete “could” or change with may

Changed, thank you.

Other examples of incorrect or obscure sentences:

Line 302: change and clarify the meaning of “inhibition against the pathogenesis”

In figure 1 legend, the description has been altered to, “Schematic representation of the relevant signaling pathway probably involved in the pathogenesis of polycystic kidney disease (PKD) and/or chronic kidney disease (CKD).”

Lines 336-337: clarify the meaning of “If that is the case, “engram” modulation with the modification of gut microbiome might be beneficial for the superior treatment tactic”

According to this suggestion, the explanation has been expanded about the“engram”.

All the figures need to be extensively modified because they are too generic, unclear and poorly detailed both in the diagrams and in the legends and what is "omitted for clarity" does not actually make the figure understandable.

Containing the non-relevant matter (signaling pathway) to figures, it would be looked more chaotic and messy, we supposed. However, the sentence including "omitted for clarity" has been deleted from the legend of Figure 2. In addition, all the figures have been amended more understandable as much as possible for the readers.

Comments on the Quality of English Language

Extensive editing of English language required

Again we have gone over the text/abstract and amended typos and grammatical errors as much as possible to improve the manuscript.

Reviewer 3 Report

Comments and Suggestions for Authors

In the manuscript: “Mitophagy and redox balance involved in the development of polycystic kidney and renal dysfunction.” The authors reviewed polycystic kidney disease, a chronic kidney disease associated with cellular processes such as mitophagy, hypoxia, AMPK, and even gut microbiome. It is an interesting topic that contributes to knowledge in the area, but certain issues must be corrected.

Major revisions

1.    In the abstract and introduction, the objective of the manuscript must be mentioned, mentioning the gap that the manuscript will fill within the current knowledge, the topics that will be reviewed, and the possible conclusions that the reader will find through the manuscript.

2.    The authors only touch on the topic of two kidney diseases associated with autophagy/mitophagy in section 3, so they are encouraged to write about more kidney diseases, such as ischemia/reperfusion (I/R) or unilateral ureteral obstruction (UOO).

3.    The authors are urged to make a table mentioning the treatments that regulate autophagy/mitophagy for treating the kidney diseases mentioned in the manuscript, which includes PKD and the others mentioned and even the new ones added, such as I/R or UOO.

Comments on the Quality of English Language

no comments

Author Response

For Reviewer3

In the manuscript: “Mitophagy and redox balance involved in the development of polycystic kidney and renal dysfunction.” The authors reviewed polycystic kidney disease, a chronic kidney disease associated with cellular processes such as mitophagy, hypoxia, AMPK, and even gut microbiome. It is an interesting topic that contributes to knowledge in the area, but certain issues must be corrected.

Major revisions

  1. In the abstract and introduction, the objective of the manuscript must be mentioned, mentioning the gap that the manuscript will fill within the current knowledge, the topics that will be reviewed, and the possible conclusions that the reader will find through the manuscript.

According to this suggestion, the title of the manuscript, abstract and introduction have been improved.

  1. The authors only touch on the topic of two kidney diseases associated with autophagy/mitophagy in section 3, so they are encouraged to write about more kidney diseases, such as ischemia/reperfusion (I/R) or unilateral ureteral obstruction (UOO).

I'm sorry I didn't make it clear enough. The purpose of this manuscript is for the development of treatment against polycystic kidney disease. Therefore, again, the title of the manuscript, abstract and introduction have been improved. It would need vast space to describe about various kidney diseases.

  1. The authors are urged to make a table mentioning the treatments that regulate autophagy/mitophagy for treating the kidney diseases mentioned in the manuscript, which includes PKD and the others mentioned and even the new ones added, such as I/R or UOO.

That is very good idea in case handling with the various kidney diseases. We would like to write a paper like that in the near future. Thank you so much.

Round 2

Reviewer 2 Report

Comments and Suggestions for Authors

The revision was done adequately by improving the review.

The manuscript may be accepted for publication in its present form.

Round 2

Reviewer 2

The revision was done adequately by improving the review. The manuscript may be accepted for publication in its present form.

Thank you so much for the good evaluation to our revised manuscript.

Reviewer 3 Report

Comments and Suggestions for Authors

I understand the meaning of the manuscript but the authors have a very general subtitle (3. Autophagy/mitophagy in various kidney diseases) that does not specify that this section will only be directed to "polycystic kidney disease" or the authors decide to change the subtitle or write on mitophagy in various kidney diseases. The above to avoid confusion in the reader.

On the other hand, it seems to me that the authors, through their manuscript, talk about many treatments associated with polycystic kidney disease, even if they do not want to talk about other kidney diseases associated with autophagy/mitophagy, so they could well make a table that summarizes these treatments.

Comments on the Quality of English Language

No comments

Round 2

Reviewer 3

I understand the meaning of the manuscript but the authors have a very general subtitle (3. Autophagy/mitophagy in various kidney diseases) that does not specify that this section will only be directed to "polycystic kidney disease" or the authors decide to change the subtitle or write on mitophagy in various kidney diseases. The above to avoid confusion in the reader.

According to this suggestion, the subtitle of section 3 has been altered.

  1. Autophagy/mitophagy involved in the pathogenesis of several kidney diseases including polycystic kidney

On the other hand, it seems to me that the authors, through their manuscript, talk about many treatments associated with polycystic kidney disease, even if they do not want to talk about other kidney diseases associated with autophagy/mitophagy, so they could well make a table that summarizes these treatments.

Although we do talk about other kidney diseases associated with autophagy/mitophagy, there are insufficient papers showing substantial evidence for the treatment. Please let me take the issue for the next opportunity of our paper in the near future. However, the figure 3 has been additionally improved according to this suggestion.